# A Comprehensive Instrument to Measure Teachers’ Attitude towards Quality Management in the Context of Online Education

**DOI:** 10.3390/ijerph19031168

**Published:** 2022-01-21

**Authors:** Katerina Tzafilkou, Liliana Mâță, Gabriela Livia Curpănaru, Ionuț Viorel Stoica, Lucian Nicolae Voinea, Constantin Șufaru

**Affiliations:** 1Information Systems IPPS, University of Macedonia, 54006 Thessaloniki, Greece; tzafilkou@uom.edu.gr; 2Teacher Training Department, Vasile Alecsandri University of Bacau, 600115 Bacau, Romania; 3Engineering and Management Department, Gheorghe Asachi Technical University of Iasi, 600115 Iasi, Romania; gabriela.curpanaru@gmail.com; 4Department of Biology, Ecology and Environmental Protection, Vasile Alecsandri University of Bacau, 600115 Bacau, Romania; 5Department of Physical Education and Sports Performance, Vasile Alecsandri University of Bacau, 600115 Bacau, Romania; lucian.voinea@ub.ro (L.N.V.); sufaruconstantin@ub.ro (C.Ș.)

**Keywords:** education quality, online education, quality management scale, teacher attitude

## Abstract

The purpose of this study is to elaborate and validate a scale for the evaluation of the teachers’ attitude towards quality management, by integrating elements of online education. Nine hundred and forty-two teachers from Romania have participated in the study. The exploratory factor analysis has led to the identification of three main dimensions of the scale: (1) communication and alignment; (2) needs and opportunities; and (3) training and support. Teachers participating in managing positions or in quality assurance boards indicated a more positive attitude towards the three components. Furthermore, teachers of a higher teaching experience reported higher values in the needs and opportunities component. The results are useful to educational institutions, program designers and policy makers to evaluate the teachers’ attitude towards quality management.

## 1. Introduction

In today’s context of knowledge society and worldwide dynamic changes, the performance of a school organization is conditioned by the efficiency of the education quality management functioning. The measurement of the teachers’ attitude towards the manner of implementing quality management within the school organization becomes a necessary condition to improve the organizational performance. Since the system of education represents an important component of the knowledge society, it is necessary for those in charge to focus on the quality management within schools and its efficacy [1]. Quality education involves the integration of democratic values and principles and of the partners’ rights and obligations, as it is constituted out of a spirit of transparency, responsibility, and involvement from both the school as an education provider, and of the family—student and parent—as beneficiaries of the education service. To create an efficient system of quality assurance, it is necessary to introduce a system of quality management. It is a complex task which involves the change of the mentality, which is applicable to both school and university teachers, along with a wide majority of people, who are the direct beneficiaries of the reform system. Quality management represents a system of interconnected processes meant to establish a policy of quality, quality objectives and the fulfilment of a quality education [2]. The role of quality management is to provide models for the continuous development and improvement of the organizational performance, which is reflected in the increase of satisfaction among students, teachers, parents, schoolmasters, and members of the community.

The system of education quality indicators helps school managers to highlight the important fields of their own activity in relation to the advantages, disadvantages, and opportunities of development. The members of the commission, who are responsible with the quality management within the school, will analyse the strategies meant to improve the indicators in accordance with certain circumstances. According to Vlašić, Vale, and Puhar [3], the education quality indicators are held together within seven specific domains: achievements, learning and teaching, students’ support, school ethos, resources, management, leadership, and quality assurance. Several benefits of the quality management system can be highlighted within the school organization [4]: the increase in the level of the awareness and appreciation expressed by the community, along with other interested parties, the improvement of the operational efficiency, the empowerment of the teaching staff in order to identify and implement the necessary changes, the more accurate and coherent defining of the methods and responsibilities designed for performance determination, the reduction of internal costs, the improvement of management practices, the involvement and motivation of the staff for continuous progress, the introduction of the problem solving process, the identification of procedure problems and other causes of deficiency.

Changes in attitude and perception are required from all those involved, students or teachers, along with a recalibration of the position held by the society in relation to the school. Improving the quality of education is possible when the relationship between managers and teachers is based on trust [5], given that everyone believes that they have contributed to the entire decision-making process. The essential characteristics of leadership for quality management [6] refer to the correlation of individuals’ work with the organization’s objectives, ensuring a comfortable and motivating work environment, cooperation with members of the organization to improve quality, providing trust between individuals, quality performance, and leadership towards quality. A quality system should assure a philosophy of the education activity based on innovation and transformative culture [7], which will lead to a common strategic view and a style of teamwork, with a significant positive impact on the students’ work results and their personal, social, and academic development.

The impact of school closures during the COVID-19 pandemic has brought many challenges to education [8] that have affected the way quality management is implemented. Many schools have had to create specific management systems to ensure the quality of education in the context of the use of information technology. Educational institutions in both pre-university and university education have been forced to move from traditional teaching to online learning because of the COVID-19 pandemic. In this context, the challenge of managing changes in the quality of online teaching and learning has emerged. When traditional classroom teaching is replaced by online distance learning, the quality of services provided is not expected to decline [9]. According to Bates [10], quality assurance in distance education is tightly linked to the teaching and learning outcomes. Quality assurance management is harder to be applied in distance than traditional education, because of the distinct characteristics of online education considering its openness and flexible structure [11]. As a fact, investing in quality assurance is essential in the context of distance education since poor quality assurance can lead to high costs and low returns of investment [12]. Researchers agree that quality assurance in distance education should follow a systematic and continuous process, respected, and well perceived by educators and institutions [10]. Based on the above, it is a priority to maintain and adapt the attributes of service quality, such as the quality of the teaching-learning process, the quality of teaching staff, and the quality of planning, information management. Therefore, investigating teachers’ perceptions of quality assurance in the context of online education is important for effective management in schools.

Researchers agree that perceived quality education is a determinant factor to the students’ behaviour and plans and hence several scales to measure perceived quality of education have been designed [13]. Moreover, students’ perceived quality education also tends to affect their career choices and academic self-efficacy [14,15]. However, most research in the field is student-centred, while the teachers’ perceived education quality is determinant to the successfulness of the quality strategy implementation. Although there are several studies investigating the factors affecting the efficacy of quality education assurance, very few are focused on measuring the teachers’ attitudes towards education quality management. Recent studies [16,17] tend to focus on the examination of individual or professional attributes (teaching experience, individual factors, etc.) that can affect the teachers’ teaching quality in the classroom, but not in their broader quality management and assurance tasks/responsibilities. Moreover, most of the existing education quality management scales do not consider the attributes of online teaching and the management of virtual classrooms. Towards this end, this study suggests that the dimension of distance education shall be integrated in current measurements of teachers’ attitude, since the pandemic had a big socio-emotional impact on teachers and on the way that they deal with the new rising educational challenges [18,19].

Considering the above, the main research objective of this study is to propose and evaluate the Quality Management Education (QME) scale considering elements of online teaching, that were risen mainly during the pandemic. The main contribution of QME is the provision of a simple and practical instrument to measure the teachers’ attitude towards education quality management in all levels of pre-university education. The instrument integrates elements of previous measurements and introduces new items regarding the efficient management of online teaching and virtual classrooms ‘management. Towards this goal, the study also seeks to examine the role of teachers’ individual and professional factors on their attitude towards education quality management.

Based on the above, the Research Questions (RQs) are formed as follows:

RQ1: Is the suggested Quality Management Education (QME) scale valid in terms of structure, consistency, and reliability?RQ2: Are there any significant differences in the Quality Management Education (QME) constructs between different groups of teachers, including their characteristics of:(i)educational/teaching level (preschool education, primary, middle/lower secondary, secondary);(ii)teaching experience;(iii)teaching environment (rural, urban);(iv)professional/teaching degree;(v)involvement in managing position;(vi)participation in quality assurance position; and(vii)participation in the board of directors?

The findings of the study are expected to provide researchers and practitioners with a valid scale to measure the teachers’ attitude towards quality management to design strategies and approaches to engage teachers more deeply in the process of quality management in their schools. Moreover, the study results contribute towards a deeper understanding of the factors that affect teachers’ attitude towards education quality management during the pandemic.

## 2. Theoretical Background

### 2.1. Current Perspectives on Education Quality Management

Recent approaches to education quality management offer an innovative perspective on the dimensions that facilitate its successful application in school institutions. The EFQM (European Foundation for Quality Management) model is based on the principle that the staff and customer satisfaction, along with the integration in the community life, constitute the result of a correct application of the management components, such as leadership, policy and strategy, staff management, resources, and processes [20]. According to the Baldrige excellence framework, the efficacy of school management is positively influenced by seven main components [21]: leadership (governing), aiming at reaching the strategic goals of the organization by establishing a bidirectional communication bridge; strategic planning through transforming the strategic outline in an action plan; focus on the customer, targeting the level of student satisfaction in relation with the competitors and the competition comparison indicators; measurement, analysis and management of knowledge through the organization of an informational system capable of sustaining the problem prediction and decision making, a set of interconnected components; focus of the labour force, given that the majority of the team will work in direct contact with the students so as to apply the strategic planning of the schools; focus on operations and functioning, by encouraging the integration of teams and global views of the activities; and correlation between the structures of the framework. Following an analysis carried out by researchers on quality management in higher education, Tari and Dick [22] identified six main dimensions: human resource management, analysis, process management, concentration of interested parties, planning, leadership, planning, and management of providers.

Alzamil [23] came up with an integrated model of education quality management to determine the improvement of efficiency and flexibility within school institutions. Since the model takes the shape of a spiral, the established improvement actions can determine the fulfilment of a process, the continuance of the actions at the same level for a different course or the passage to a superior level. In this way, the efficiency of passing the course from one level to another has been proven in what concerns the fulfilment of all the established objectives dealing with the assurance of education quality. The new framework of the knowledge society generates new models of education quality management with the purpose of promoting a culture of quality [24] within the system of education, from the interaction between the students and the teachers, as it is carried out in traditional learning, to the liaised interaction and use of technology, which is specific to online learning.

Latif et al. [13] designed and validated a six-dimensional service quality scale, through data collected from seven different higher education institutions. Their scale composed 37 items, allocated in the dimension of teacher quality, administrative services, knowledge services, activities, continuous improvement, and leadership quality. Their model provided a useful and detailed approach; however, it did not include of items of online education and group-differences were not examined in the participating population.

Another integrated model was developed by Schijns [25] to measure the service quality in terms of online education in universities. The authors applied a PLS-SEM analysis on 1287 university students to understand the service quality factors that can affect the students’ satisfaction and intention to recommend their institution. Their findings indicated nine components, derived from the initial 12-component National Student Equity scale, that can significantly affect the students’ overall satisfaction. The new integrated model includes, within a holistic view, both internal factors of the education system (i.e., the school milieu), and external factors (Figure 1). At the basis of the integrated model, we find Deming’s wheel of quality (1993), which is a method of organizing and carrying out activities meant to continuously improve the quality management system. The phases of quality management are planning, doing, checking, and acting. A successful model of school management quality concentrates on the relations between students, teachers and the curriculum, around which several external influences gravitate: society, family, labour market requirements, need of competence and life-long learning.

### 2.2. The Attitude towards Education Quality Management

The basis of the research stems from the concept of attitude towards education quality management; however, to understand its significance, it is important to define the general concept of attitude. Eagly and Chaiken [26] defined the term ‘attitude’ as a psychological tendency expressed through the evaluation of a certain entity with a certain degree of pro and against bias. Attitudes can offer a favourable context for the understanding of the way of implementing quality management within the education system. In this context, the attitude towards education quality management represents the general evaluation of the specific activities related to the implementation of quality assurance standards. All the teachers are responsible for their own attitude concerning education quality management. Thus, the attitude towards education quality management addresses the degree of encouragement concerning the conformation to the education quality assurance standards within the school organization. The research carried out in recent years has shown an increased preoccupation for the exploration of the teachers’ attitude towards education quality management [27].

Although there is a series of studies which aim at investigating the factors which determine the efficacy of quality education assurance [21,28,29,30,31,32,33,34,35,36], very few focus on the measurement of the teachers’ attitudes towards education quality management [37] or are concerned with education quality assurance [38,39]. Some of the studies are based on the validation of questionnaires, which evaluate the teachers’ attitudes towards the dimensions of certain models, such as the total quality management [40,41]. An expansion of the research towards the identification of the teachers’ attitudes concerning organizational changes is observed, because of the introduction of Information Technology systems, of learning and teaching management in schools [42]. It was found that there are several studies focused on measuring attitudes towards the online component of quality management, especially among students [25,43,44] and very little research among teachers.

Triggered by the above, the main purpose of this study is to elaborate and validate the scale of measuring the teachers’ attitude towards the specific dimensions of quality management and examine individual and professional attributes that might affect their attitude towards education management quality.

## 3. Materials and Methods

### 3.1. Item Generation

Within this research, a questionnaire has been developed for the measurement of the teachers’ attitude towards education quality management in school education (Appendix A). The questionnaire has 42 closed-ended questions. Most of the items have been adapted after Asif et al. [45] and Menezes et al. [21], while items 6, 7, 11, 21 and 26 are original. These items were introduced to highlight the new dimension of quality management in the context of online education (the school management is deficient in carrying out online didactic activities; the virtual classroom is made available by the school; a guide with all the steps necessary for teachers and students for online didactic activities has been provided; I have been informed, by the school management, about the rules of virtual classroom handling; I have been supported by the school management in carrying out online teaching). There are also four reversed items: 6, 19, 29 and 38. The Likert scale of measurement has been used, with five potential answers, varying from 1—meaning strong disagreement to 5—strong agreement.

The items of the questionnaire have been distributed in accordance with six dimensions which are specific to education quality management: 10 items for the leadership dimension (I1, I8, I17, I27, I32, I36, I38, I40, I41, I42), 9 items for the strategic planning dimension (I2, I16, I18, I9, I15, I28, I 29, I33, I39), 7 items concerning the student-centred dimension (I3, I10, I14, I19, I24, I30, I31), 5 items referring to the employee-centred dimension (I4, I13, I20, I35, I37), 3 items for the information management dimension (I5, I12, I34) and 5 items dealing with the online teaching quality assurance dimension (I6, I7, I11, I21, I26).

### 3.2. Data Collection and Participants

A questionnaire was distributed to 967 teachers in pre-university education in Romania, to investigate the teachers’ perception on the way of implementing quality management in primary and secondary schools. All participants confirmed the approval of voluntary participation in the research. The study respects the Declaration of Helsinki concerning the rights of human subjects participating in research. The measurement scale used was the Likert Scale varying from 1 (strongly disagree) to 5 (strongly agree).

Finally, 942 teachers (815 female, 127 male) successfully completed the survey. Of them, 163 were teaching in preschool education, 222 teachers were teaching in primary education, 265 were teaching in secondary education and 292 in high school. Almost half of the teachers (*n* = 480) were teaching in urban environments and half of them (*n* = 461) in rural ones. A few of them were in a managing position (*n* = 115), while 211 teachers were members of the quality assurance commission and 245 were members of the board of directors in their schools. The majority (*n* = 558) had a high level of professional experience (didactic degree I), 99 teachers were beginner teachers, 137 teachers were definitive teachers and 148 had a didactic degree II. Finally, most of the participants (80%) were using the platform of Google Classroom, the rest were using Microsoft Teams and G-Suite (15%), while only a few declared that they were using Zoom or Adservio (5%).

### 3.3. Data Analysis

This study applied a Partial Least Square Stractural Equation Modeling (PLS-SEM) approach using SmartPLS software [SmartPLS GmbH, Bönningstedt, Germany] to measure and validate the suggested education quality management scale. According to Bentler and Huang [46] and Dijkstra and Henseler [47] PLS-SEM can consistently mimic common Covariance-based SEM (CB-SEM) approaches. Moreover, researchers support that it is more suitable for complex models and social science and exploratory research [48,49], and similar research applied the PLS-SEM approach as the key Confirmatory Factor Analysis (CFA) method [50]. On the other side, a CB-SEM approach should be chosen if “the goal is theory testing, theory confirmation or comparison of alternative theories” [49] (p.144). Although many researchers focus on comparing the differences of model estimations when using CB-SEM and PLS-SEM, both methods are complementary rather than competitive. Furthermore, PLS-SEM was chosen as the CFA method because of the non-normal distribution on the data based on their values of skewness (<3.0) [51] and normality test (*p* < 0.005) [52].

To extract the dimensional structure of the suggested QME model, Exploratory Factor Analysis (EFA) is conducted on the defined set of items. To confirm and establish the structural validity of the scale, a PLS-SEM CFA is conducted on the EFA extracted components. The final model is evaluated in terms of model fitness, internal consistency, composite reliability, convergence validity and discriminant validity.

To examine the significant differences in QME components across different groups of teachers, non-parametric statistical methods (Mann–Whitney and Kruskal–Wallis) were applied because of the non-normal distribution of the data [52].

The Exploratory Factor Analysis (EFA) and the statistical analyses of descriptive statistics and tests for significant differences between groups were performed through SPSS software. The PLS-SEM analysis and evaluation of the model was applied in SmartPLS software.

## 4. Results

### 4.1. Suitability of Data for Factor Analysis

The Barlett’s test of sphericity and the Kaiser–Meyer–Olkin (KMO) test were conducted to investigate the factorability of the data and the adequacy of the sample. Results indicated a significant test statistic for Bartlett’s test of *p* < 0.001, and a high KMO value (Table 1), confirming the suitability of the data for structural analysis. Moreover, the Spearman correlation analysis indicated relatively high correlations among items, confirming their suitability for factor analysis.

### 4.2. Exploratory Factor Analysis

The dimensional structure of the QME scale was identified through exploratory factor analysis (EFA). All 42 items were considered in the first round EFA. EFA was conducted using the principal axis factoring method and a Varimax rotation. All items of communality scores lower than 0.4 were excluded and then a second EFA round was performed. The second EFA indicated three components with eigenvalues above 1.0. All items performing lower than a 0.5 factor load [53] were removed. Some of the newly embedded items of online teaching were removed due to low factor loadings (<0.5). These items included: (i) The school management is deficient in carrying out online didactic activities; and (ii) A guide with all the steps necessary for teachers and students for online didactic activities has been provided. The final three factor model included 20 items and accounted for 67% of the total variance, as depicted in Table 2.

The first dimension called “Communication and Alignment” and was composed of ten items (Cronbach’s alpha = 0.940), including two items of online teaching ((i) The virtual classroom is made available by the school; (ii) I have been informed, by the school management, about the rules of virtual classroom handling). The second dimension called “Needs and Opportunities” and was composed of six items (Cronbach’s alpha = 0.895). The third dimension called “Training and Support” and was composed of four items (Cronbach’s alpha = 0.871), including one item of online teaching ((i) I have been supported by the school management in carrying out online teaching).

### 4.3. Confirmatory Factor Analysis and Model Fit

A PLS-SEM CFA was conducted on the 20 items extracted by EFA, through SmartPLS software to confirm and establish the structural validity of the scale.

The resulted model fit indices (Chi-Square = 1663.676, NFI = 0.893, SRMR = 0.050) indicated a good fit between the model and the observed data [54,55,56]. Moreover, the scores of the outer loading factors were valid (>0.7). The model’s internal consistency was evaluated in terms of Rho_alpha and composite reliability (CR). All the values of Cronbach alpha and Composite Reliability (CR) demonstrated internal consistency (>0.7) [47] and all AVE values were above 0.5 [57,58,59]. Item-total correlation were also examined, and significant correlations were shown to exist between the factors (*p* < 0.01).

Finally, the convergence validity was evaluated through Average Variance Extracted (AVE) that were all above the acceptance threshold of 0.7 [47]. The extracted values are depicted in Table 3.

### 4.4. Discriminant Validity

To reinforce the validity of the construct validity, the discriminant validity was assessed according to the criterion of Fornel and Larcker [59], which is the most widely used method. According to this criterion, the square root of each construct’s AVE should have a greater value than the correlations with other latent constructs. As depicted in Table 4, the QME scale supports the discriminant validity between the three constructs [59].

### 4.5. Descriptive Statistics

The descriptive statistics results indicated that teachers hold average a positive attitude towards the three components of the QME scale. As depicted in Table 5, the component of communication and alignment received the highest scores, compared to the rest components of needs and opportunities, and training and support. It is also worth mentioning that all items of online teaching received high scores (>3.9) (CAL2: mean = 4.35, stdev = 1.03, CAL5: mean = 4.13; stdev = 1.16; TRS3: mean = 3.95; stdev = 1.27), indicating a positive attitude towards the online teaching dimensions of quality education in the examined teacher population.

### 4.6. Differences between Teacher Groups

The normality distribution test indicated that the factor items are not normally distributed (*p* < 0.01) in the examined groups of teachers. Hence, non-parametrical statistical methods (Mann–Whitney and Kruskal–Wallis) were used to examine the potential significant differences in QEMS within the examined teacher groups.

The teacher groups were defined from the questionnaire collected feedback, according to the following attributes:Educational/teaching level: preschool, primary, middle/lower secondary, secondary;Teaching environment (area where the school institution is located): urban, rural;Professional/teaching degree (beginning teacher, definite teacher, teacher with didactic degree II, teacher with didactic degree II);Teaching experience (in years);Managing position: yes, no;Member of the Commission for Quality Evaluation and Assurance (or a similar entity within the school): yes, no;Member of the Council of Administration (board of directors, etc.): yes, no.

Gender was excluded from the analysis, since female teachers out-weighed males in the sample. Similarly, we did not examine any differences based on the used e-learning platform, since the majority (80%) reported the use of the same platform, Google Classroom.

Educational/teaching level and teaching environment did not indicate any association with the scale components. Teaching experience reveled significant differences in the component of NOP where teachers with didactic degree I reported the highest values in the component, while teachers with didactic degree II reported the lowest values.

Interestingly, all the teacher groups related to managing roles or positions revealed significantly higher scores across all the QEM constructs. Teachers who held a managing position perceived higher items in the three components compared to teachers who were not occupied in any managing position. Likewise, the teachers who were members of the quality assurance team or were on the directory board reported significantly higher values in the measured components (CAL, NOP, TRS) than those teachers who were not members. The effect sizes of the detected significant differences were examined based on the epsilon squared value, and the result was interpreted based on the standardized difference as suggested in Rosenthal [60] (p. 19). The results indicated medium effects (0.2 < r ≤ 0.5) in the CAL and TRS components between different groups of managing positions, in TRS between different groups of members of the broad directors, and in all three components (CAL, NOP, TRS) between different groups of teaching experience. The calculated effect sizes where low (r ≤ 0.2) in the other cases, implying that future research should be conducted in different teacher populations, or a replicated study, as suggested in [61].

All significant associations are presented in Table 6.

## 5. Discussion and Implications

The main purpose of this study was to propose and validate a new scale to measure the teachers’ attitude towards quality management in education, across all pre-university stages. The suggested model was based on previous scales encompassing items leadership, strategic planning, student-centred items, employee-centred items, information management, online teaching quality assurance dimension as well as several original items suggested in the study. The main difference with previous scales is the integration of all quality management dimensions in a simple scale and the adjustment to the current trends of online teaching and learning caused by the pandemic. Hence, the proposed scale includes elements with regards to the provision of school support on online teaching activities and on the management of virtual classrooms.

The EFA results generated a new three-dimensional model, clearly indicating the components of communication and alignment, needs and opportunities, and training and support. Regarding communication and alignment, it is more than obvious that communication practices and technologies have become increasingly important for school organizations. Adequate internal communication within the educational institution has had a positive impact on organizational effectiveness and efficiency. In terms of the needs and opportunities that define the quality management, it is important that educational services comply with the requirements of the main beneficiaries. Therefore, the quality management is the expression of the usefulness of the product offered, as well as the extent to which by all its characteristics meets the needs of students, teachers, parents and society. From the perspective of training and support, it is certain that the quality of teachers and managers depend on the teaching and learning processes and the results of education. Continuous professional training of teachers in new fields through training, counselling, and consulting programs is a strategic direction of quality management.

The PLS-SEM results revealed the validation of the scale, indicating internal validity, reliability, convergence validity and model fitness. The analysis of differences between the examined teachers’ groups revealed interesting insights. The findings come in accordance with previous studies in the field of traditional and face-to-face education. As the results of teaching experience (in years) was positively associated with the teachers’ attitude towards QEM. This finding comes in accordance with previous studies, where teachers of a higher teaching experience tend to indicate more positive attitudes towards teaching trends, e.g., online teaching [62,63] or digital integration [64]. Previous studies have also proved a positive relationship between the teachers’ teaching experience with the quality of teaching [16]. According to the results of the research conducted by Elumalai et al. [43], there is a positive relationship between different variables and the quality of e-learning in higher education. The results of recent studies [44] have shown that students’ attitudes towards e-learning are positively influenced by certain factors, such as the perceived usefulness of e-learning, self-management of learning and self-efficacy. The data of another research [25] show that the most important factors of students’ satisfaction with the quality of services at the level of online education in universities are the content and structure of the study and Professors/Lecturers, followed by academic guidance and counseling, testing and evaluation and the task of study. Similarly, teachers holding managing and administrative roles or positions reported significantly higher scores in the QEM dimensions. This finding highlights the need to further engage teachers with managerial tasks, to leverage their interest and positive attitude towards quality management in education.

Overall, the study resulted in the construction of a an up-to-date, comprehensive, and reliable questionnaire focused on education quality management that can be used in future studies to draw useful conclusions on the teachers’ attitude towards QEM and to identify the factors determining quality management and quality assurance in education. Theoretically, the findings of this study offer additional insights to researchers in understanding the factors that affect the teachers’ perceived quality education, as well as further details on perceived quality items regarding online teaching and virtual classroom management. The replication and evaluation of this model in different populations can contribute to the deeper understanding of the role of individual attributes on teachers’ perceived quality education in different educational contexts and countries.

Moreover, the proposed Quality Management Education Scale provides a fast and practical instrument that can be applied by educational institutions and professional development program designers to evaluate the teachers’ attitude towards quality management, across all pre-university levels. Professional development program designers and educational institutions can apply the scale to better design the teachers’ development paths and strategies to engage them in quality management tasks.

## 6. Limitations

This study brings some limitations. One main limitation is the generalizability of the findings. The participants come from one country (Romania) and belong in the pre-university teaching stages. Similar studies in different populations might generate different results. One other limitation is the underrepresentation of male teachers in the examined sample. Although gender has been excluded from the group-based analysis, it might have affected the differences indicated in different groups of teachers where there were significantly more female teachers than male teachers who tend to share similar attitudes. Finally, the survey is based on individual self-reported measures and hence it is prone to bias. Future research should extend this wok by using different methods of data collection, like, for instance, observations, course recordings, and focus groups.

## 7. Conclusions

The efficient implementation of education quality management offers benefits both to the students and to teachers, and to society; it generates the orientation of the education process and its lining towards standards, its continuous improvement and responsibility. This solid basis is applicable to schools and higher education, along with lifelong learning. In the long run, quality management contributes to sustainable economic growth and to the formation of more stable and responsible governments, precisely because it assures the adaptation to the requirements, needs and learning styles of the students, to the current values of the society, as well as to future perspectives.

To this end, this study designed and validated an instrument to efficiently assess the teachers’ attitude towards quality management of education, across all pre-university teaching stages. The proposed scale is composed of 20 items and three components: (1) communication and alignment; (2) needs and opportunities; and (3) training and support. The PLS-SEM-based CFA demonstrated the scale’s validity and reliability, indicating internal consistency, convergence validity and model fitness. The examination of a series of individual and professional (managerial) attributes revealed several significant differences between different groups of teachers. The findings of this study shed light on the role of individual factors on teachers’ attitude towards the education quality management and provide a valid and practical scale that can be implemented across all levels of pre-university education. The results are useful to educational institutions, program designers and policy makers to evaluate the teachers’ attitude towards quality management and design strategies to engage teachers in quality management tasks and to achieve efficient quality management outcomes.

## Figures and Tables

**Figure 1 ijerph-19-01168-f001:**
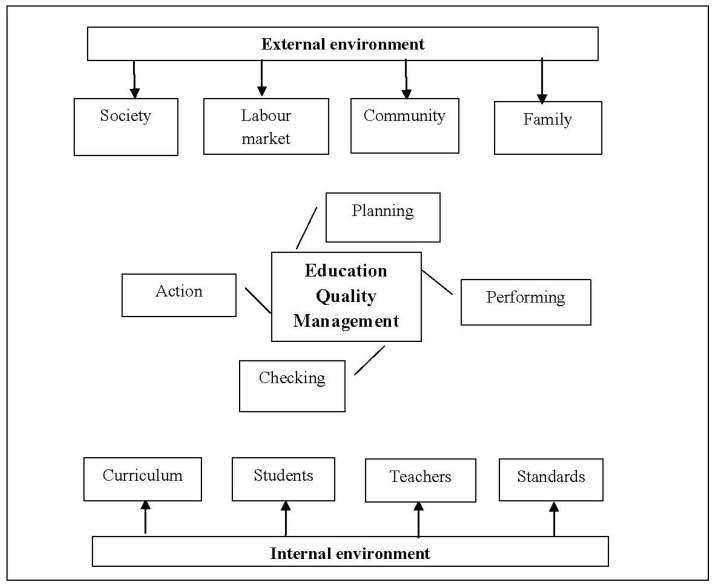
The integrated model of education quality management.

**Table 1 ijerph-19-01168-t001:** KMO and Bartlett’s Test.

Kaiser–Meyer–Olkin Measure of Sampling Adequacy		0.982
Bartlett’s Test of Sphericity	Approx. Chi-Square	32,455.276
	df	741
	Sig.	0.000

**Table 2 ijerph-19-01168-t002:** Results of EFA of the 18—item perceived Quality Management Education Scale (QMES).

Factor	Item ^1^	Factor Loading	Eigen Value	Cumulative Variance	Rotation Sums of Squared Loadings ^2^
CAL	Communication and Alignment		21.732	61.330	19.223
CAL1	The school principal has knowledge concerning the system of quality management and implementing.	0.741			
CAL2 *	The virtual classroom is made available by the school.	0.874			
CAL3	The school organization collects and operates statistical data (e.g., student record, class attendance) in order to improve the education process.	0.671			
CAL4	Within the school, there are initiatives of promoting honest and direct communication.	0.522			
CAL5 *	I have been informed, by the school management, about the rules of virtual classroom handling.	0.539			
CAL6	The managing staff communicates efficiently with every person from the institution.	0.571			
CAL7	The managing staff is well acquainted with the concepts related to quality and with the new competences needed for the application of the quality management system.	0.733			
CAL8	The actions of the school management are in accordance with the mission, vision, and values of the organization.	0.578			
CAL9	The actions of the managing staff show their ethical commitment and respect for the law.	0.839			
CAL10	The school management concentrates on improving student and staff performance.	0.642			
NOP	Needs and Opportunities		2.000	64.325	17.985
NOP1	The needs and suggestions of the business environment are considered when designing the curriculum.	0.628			
NOP2	The system of assuming complaints, suggestions, criticism, and appreciation offers quick measures for problem solving.	0.520			
NOP3	The students’ requests are considered when designing elective disciplines.	0.578			
NOP4	Within the school, there are attractive, stimulating programmes meant to bring new students.	0.619			
NOP5	The organization benefits from the opportunities of innovation in educational services.	0.629			
NOP6	The students are involved in solving the problems found.	0.612			
TRS	Training and Support		1.093	67.122	18.222
TRS1	The school management provides adequate resources for didactic and administrative staff training.	0.739			
TRS2	Information on training programmes is given by the school.	0.510			
TRS3 *	I have been supported by the school management in carrying out online teaching.	0.796			
TRS4	The school delivers surveys concerning the employees’ workplace satisfaction.	0.663			

^1^ All the items are measured on a 5-point Likert scale (1: strongly disagree to 5: strongly agree). ^2^ When factors are correlated, sums of squared loadings cannot be added to obtain a total variance. * New items embedded regarding online teaching and virtual classroom.

**Table 3 ijerph-19-01168-t003:** Construct reliability and validity of the Quality Management Education scale (QMES).

Cronbach’s Alpha	rho_A	Composite Reliability	Average Variance Extracted (AVE)
CAL	0.940	0.950	0.658
NOP	0.895	0.897	0.656
TRS	0.871	0.872	0.721

**Table 4 ijerph-19-01168-t004:** Discriminant validity.

Latent Constructs	CAL	NOP	TRS
CAL	0.811		
NOP	0.849	0.810	
TRS	0.837	0.782	0.849

Note. The discriminant validity was assessed using the criteria of Fornel and Larcker [59] by comparing the square root of each AVE in the diagonal with the correlation coefficients (off-diagonal) for each construct in the relevant rows and columns.

**Table 5 ijerph-19-01168-t005:** Descriptive statistics (*n* = 942).

	Minimum	Maximum	Mean	Std. Deviation
CAL	1.00	5.00	4.197	0.914
NOP	1.00	5.00	3.738	0.936
TRS	1.00	5.00	3.723	1.085

**Table 6 ijerph-19-01168-t006:** Significant differences in the Management of Quality of Education component results between teacher groups (Mann–Whitney and Kruskal–Wallis test).

	CAL	NOP	TRS
Grouping Variable: Managing position			
Mann-Whitney U	34,934.500	38,195.500	30,862.500
Wilcoxon W	377,312.500	380,573.500	373,240.500
Z	−4.662	−3.429	−6.131
Asymp. Sig. (2-tailed)	0.000 *	0.001 *	0.000 *
Grouping Variable: Member of the quality assurance commission			
Mann-Whitney U	67,175.500	66,565.500	65,303.500
Wilcoxon W	334,721.500	334,111.500	332,849.500
Z	−2.885	−3.037	−3.409
Asymp. Sig. (2-tailed)	0.004 *	0.002 *	0.001 *
Grouping Variable: Member of the board of directors			
Mann-Whitney U	69,822.500	70,167.000	64,053.000
Wilcoxon W	313,075.500	313,420.000	307,306.000
Z	−4.290	−4.161	−5.847
Asymp. Sig. (2-tailed)	0.000 *	0.000 *	0.000 *
Grouping Variable: Teaching experience			
Chi-Square	7.296	12.199	4.871
df	3	3	3
Asymp. Sig.	0.063	0.007 *	0.182

* Statistical significance at level *p* = 0.05.

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
