# Peer review of "A Comprehensive Instrument to Measure Teachers’ Attitude towards Quality Management in the Context of Online Education"

_ijerph, 2022, doi:10.3390/ijerph19031168_

Round 1
Reviewer 1 Report
Mainly, I have minor remarks that I would ask the authors to revise:
(1) I miss an explicit presentation of the research questions. These questions should indicate distinctly when the developed scale should be considered as valid.
(2) n my opinion, Covid-19 and distance learning are dealt with too briefly in the introduction and in the theoretical background. Here, the authors should describe in more detail the evidence that is already available with regard to their research interest.
(3) The data analysis chapter should be more detailed. Many deteails that only become clear in the results chapter are not included in this chapter. In addition, I think the authors should avoid using acronyms such as CB-SEM, since these are often confusing for readers.
(4) Table 4 needs a more clear labeling. It is not clear which parameters are shown in the rows or columns of the table.
(6) In chapter 4.6, the authors should report the effect sizes of the group differences (e.g., in table 6). Statistical significance does not imply statistical relevance.
Author Response
Comment#1: I miss an explicit presentation of the research questions. These questions should indicate distinctly when the developed scale should be considered as valid.
Reply: First, we would like to thank the reviewer for this suggestion. To explicitly define ethe research questions we have integrated in the Introduction section the following part:” Based on the above, the Research Questions (RQs) are formed as follows:
RQ1: Is the suggested Quality Management Education (QME) scale valid in terms of structure, consistency, and reliability?
RQ2: Are there any significant differences in the Quality Management Education (QME) constructs between different groups of teachers, including their characteristics of:
- educational/ teaching level (preschool education, primary, middle/lower secondary, secondary),
- teaching experience,
- teaching environment (rural, urban),
- professional/ teaching degree,
- involvement in managing position,
- participation in quality assurance position, and
- participation in the board of directors?”.
Comment#2: In my opinion, Covid-19 and distance learning are dealt with too briefly in the introduction and in the theoretical background. Here, the authors should describe in more detail the evidence that is already available with regard to their research interest.
Reply: As well suggested, we have added enough evidence in Introduction to describe the importance of quality assurance in the context of online learning as emerged by covid-19. The added test of the following: “Educational institutions in both pre-university and university education have been forced to move from traditional teaching to online learning as a result of the COVID-19 pandemic. In this context, the challenge of managing changes in the quality of online teaching and learning has emerged. When traditional classroom teaching is replaced by online distance learning, the quality of services provided is not expected to decline (Istijanto, 2021). According to Bates (2019), quality assurance in distance education is tightly linked to the teaching and learning outcomes. Quality assurance management is harder to be applied in distance than traditional education, because of the distinct characteristics of online education considering its openness and flexible structure (Buthcher & Hossen, 2014). As a fact, investing in quality assurance is essential in the context of distance education, since poor quality assurance can lead to high costs and low returns of investment (Jung & Latchem, 2012). Researchers agree that quality assurance in distance education should follow a systematic and continuous process, respected, and well perceived by educators and institutions (Bates, 2019). Based on the above, it is a priority to maintain and adapt the attributes of service quality, such as the quality of the teaching-learning process, the quality of teaching staff, the quality of planning, information management. Therefore, investigating teachers' perceptions of quality assurance in the context of online education is important for effective management in schools.”
Comment#3: The data analysis chapter should be more detailed. Many deteails that only become clear in the results chapter are not included in this chapter. In addition, I think the authors should avoid using acronyms such as CB-SEM, since these are often confusing for readers.
Reply: We thank the reviewers for this useful suggestion. As suggested, we have added the explanation of the statistical acronyms in the text: “…mimic common Covariance-based SEM (CB-SEM) approaches”. “… Confirmatory Factor Analysis (CFA) method….”.
Moreover, we enriched the dada analysis section by providing further details on the methodological steps that we followed to build and validate the model. Excerpt of the added text is provided below:
“To extract the dimensional structure of the suggested QME model, Exploratory Factor Analysis (EFA) is conducted on the defined set of items. To confirm and establish the structural validity of the scale a PLS-SEM CFA is conducted on the EFA extracted components, The final model is evaluated in terms of model fitness, internal consistency, composite reliability, convergence validity and discriminant validity.
To examine the significant differences in QME components across different groups of teachers, non-parametric statistical methods (Mann Whitney and Kruskal Wallis) were applied because of the non-normal distribution of the data [48].
……
The PLS-SEM analysis and evaluation of the model was applied in SmartPLS software.”.
Comment#4: Table 4 needs a more clear labeling. It is not clear which parameters are shown in the rows or columns of the table.
To clarify and interpret the depicted values in Table 4, we added a table-heading named “Latent constructs”, as well as a Note below the Table explaining the following: The discriminant validity was assessed using Fornel and Larcker [55] by comparing the square root of each AVE in the diagonal with the correlation coefficients (off-diagonal) for each construct in the relevant rows and columns.
Comment#5: In chapter 4.6, the authors should report the effect sizes of the group differences (e.g., in table 6). Statistical significance does not imply statistical relevance.
Reply: We thank the reviewer for this useful observation. As suggested, we calculated the effect size of the detected differences, based on the formula suggested for non-parametric statistics (r=Z/SQRT(N), Cohen (1988). To explain this to the readers we added the following text, before Table 6: “The effect sizes of the detected significant differences were examined based on the epsilon squared value, and the result was interpreted based on the standardized difference (Cohen’s d) as suggested in Cohen (1988). The results indicated medium effects (0.2>d<0.5) in the CAL and TRS components between different groups of managing positions, in TRS between different groups of members of the broad directors, and in all three components (CAL, NOP, TRS) between different groups of teaching experience. The calculated effect sizes where low (d ≤ 0.2) in the rest cases, implying that future research should be conducted in different teacher populations, or a replicated study as suggested in Funder and Ozer (2019).”

Reviewer 2 Report
The paper has very interesting ideas about online education. However it is not well presented. First there are too many long sentences that are hard to understand and need to be broken up into shorter sentences. Second I would recommend the authors to compare these statistics relative to traditional face to face teaching and learning environment.

Author Response
In the Discussion section we have added some text to clarify that we discuss previous findings in the context of online and traditional education, compared to ours. For this, we have added the following phrase “The findings come in accordance with previous studies in the field of traditional and face-to-face education.”. The rest of the paragraph describes and compares previous findings: “As the results teaching experience (in years) was positively associated with the teachers’ attitude towards QEM. This finding comes in accordance with previous studies, where teachers…………..”.

Reviewer 3 Report
Dear authors the paper is well written and well organized. The methodology, results, disccussion and conclusions are also correct.
Author Response
We thank the reviewer for approving and appreciating our work.

Round 2
Reviewer 1 Report
From my point of view, the manuscript is almost ready to be accepted. I just discovered one more error in the manuscript, which should be fixed: The effect size measure the authors use is Rosenthal's r and not Cohen's d (see Rosenthal, 1991, p. 19). In addition, corrections should be made in lines 403 and 407:
- Line 403: "(0.2 < r ≤ 0.5)" instead of "(0.2 > d < 0.5)".
- Line 407: "(r ≤ 0.2)" instead of "(d ≤ 0.2)".
Rosenthal, R. (1991). Meta-analytic procedures for social research. London, Sage Publications Ltd.
Author Response
The changes have been made. Thanks so much for the directions!
Line 403: "(0.2 < r ≤ 0.5)"
Line 407: "(r ≤ 0.2)"
